# Satisfying QTPP of Erythropoietin Biosimilar by QbD through DoE-Derived Downstream Process Engineering

**DOI:** 10.3390/pharmaceutics15082087

**Published:** 2023-08-04

**Authors:** Kakon Nag, Enamul Haq Sarker, Samir Kumar, Sourav Chakraborty, Maksusdur Rahman Khan, Mashfiqur Rahman Chowdhury, Rony Roy, Ratan Roy, Bipul Kumar Biswas, Emrul Hasan Bappi, Mohammad Mohiuddin, Naznin Sultana

**Affiliations:** Globe Biotech Limited, 3/Ka (New), Tejgaon I/A, Dhaka 1208, Bangladesh

**Keywords:** erythropoietin, biosimilar, critical process parameter (CPP), design of experiment (DoE), purification, quality by design (QbD), quality target product profile (QTPP), downstream, process engineering, validation

## Abstract

Well-characterized and scalable downstream processes for the purification of biologics are extremely demanding for delivering quality therapeutics to patients at a reasonable price. Erythropoietin (EPO) is a blockbuster biologic with diverse clinical applications, but its application is limited to financially well-off societies due to its high price. The high price of EPO is associated with the technical difficulties related to the purification challenge to obtain qualified products with a cost-effective defined process. Though there are reports for the purification of EPO there is no report of a well-characterized downstream process with critical process parameters (CPPs) that can deliver EPO consistently satisfying the quality target product profile (QTPP), which is a critical regulatory requirement. To advance the field, we applied the quality by design (QbD) principle and design of experiment (DoE) protocol to establish an effective process, which is scalable up to 100× batch size satisfying QTPP. We have successfully transformed the process from static mode to dynamic mode and validated it. Insignificant variation (*p* > 0.05) within and between 1×, 10×, and 100× batches showed that the process is reproducible and seamlessly scalable. The biochemical analysis along with the biofunctionality data ensures that the products from different scale batches were indifferent and comparable to a reference product. Our study thereby established a robust and scalable downstream process of EPO biosimilar satisfying QTPP. The technological scheme presented here can speed up the production of not only EPO but also many other life-saving biologics and make them available to the mass population at a reduced cost.

## 1. Introduction

Erythropoietin (EPO) is a glycoprotein cytokine, also known as hematopoietin or hemopoietin [1]. This biomolecule is involved in the differentiation, proliferation, and maintenance of the physiological levels of erythroid stem cells [2,3]. In the adult stage, it is produced by interstitial fibroblasts in the kidney; it is also produced in perisinusoidal cells in the liver during the fetal and perinatal periods. Recombinant human erythropoietin (rhEPO) is a biotechnologically produced therapeutic protein. EPO and rhEPO are used interchangeably in this article hereafter. EPO is composed of 165 amino acids glycoprotein and its estimated molecular weight is 34 kDa [4]. It has 5–8 isomeric forms over the isoelectric point (pI) range of 4.4–5.2 [5]. There are three potential N-linked glycosylation sites on Asn24, Asn38, Asn83, and 1 O-glycosylation site on Ser126 [6,7,8,9]. These glycosylations cover approximately 40% molecular weight of the protein. EPO has two disulfide bonds between the cysteines 7–161 and 29–33, which are essential for maintaining biological activity [4,10]. Functional EPO cooperates with various other growth factors such as interleukin (IL)-3, IL-6, glucocorticoids, and stem cell factor (SCF) to develop erythroid lineage from multipotent progenitors. EPO binds to the EPO receptor (EPOR) on the surface of the RBC progenitor and activates the JAK2 signaling cascade [11,12]. This initiates the STAT5, PIK3, and Ras MAPK pathways, which results in differentiation, survival, and proliferation of the erythroid cell [13].

EPO has been used to treat anemia related to chronic kidney disease (CKD), cancer chemotherapies, zidovudine patients with HIV infection, etc. [14]. It has been reported that rhEPO is potent for ribavirin treatment in patients with hepatitis C [15]. CKD is a rising global health problem, and anemia is a serious complication of CKD that has significant adverse outcomes [16]. There are more than 850 million people globally who suffer from CKD, which is increasing every year [17]. More than 37 million American adults may have CKD, and it is estimated that more than 1 out of every 7 people with kidney disease have anemia [18]. It was also reported that the study-based estimated prevalence of anemia was 15% among CKD patients in the USA, 45–55% among Asian CKD patients, and 50–90% among African CKD patients [19].

The rising prevalence of anemia due to CKD has been driving the global demand for EPO drugs. In 2019, the global EPO drug market was expected to grow more than USD 18.67 billion by 2025 at a compound annual growth rate (CAGR) of 11.65% during the forecast period (2018–2025) [20]. Raising demand and limited supply of qualified rhEPO lead to the treatment of anemia and other blood deficiency diseases expensive. The cost of treatment of anemia by rhEPO was estimated at USD 24,128.03 for the Hb level 9–10 g/dL and USD 28,022.33 for the Hb level 11–12 g/dL per quality-adjusted life-year [21]. In 2012, Matti et al. reported that the weight-based dosing cost of biosimilar EPO treatment was EUR 5484 once a week, and it increases up to EUR 7168 [22]. According to a recent report by World Bank, the global average GDP per capita was USD 11,570 [23]. Ironically, the average GDP of the global population is below the average medication cost of EPO treatment. Further, the cost of EPO treatment is creeping up steadily due to the growing concerns for regulatory issues associated with the satisfying of the finer level of the quality target product profile (QTPP). Therefore, it is an ever-growing challenge to ensure quality medication at an affordable price globally.

Bioprocessing of recombinant proteins requires a complicated manufacturing process with multiple unit operations. Therefore, it is a highly rewarding task to establish a qualified process that produces cost-effective quality products. Most biotechnology unit operations are complex in nature with numerous process variables and diverse attributes of feed materials, which can have significant impacts on the performance of the process and the QTPP of the final product. The diversity of biotherapeutics and protein expression systems demonstrated the necessity of competent process development that can increase the cost-effective productivity at the commercial scale [24]. Quality by design (QbD) using a design of experiments (DoE)-based approach offers suitable solutions to this conundrum and allows for efficient estimation and identification of critical process parameters (CPPs) [25]. The resultant process provides controllable operational conditions for obtaining products satisfying QTPP on a regular basis–batch after batch. Therefore, the systematic approach (Appendix A) of process development is a prominent task to achieve QTPP for each biopharmaceutical.

Though many studies are available regarding protein purification, to the best of our knowledge, there are none with the definitive process characterization indicating CPPs, and controllability of CPPs to achieve QTPP with the scale-up opportunity. Scalability is one of the vital factors for the protein therapeutic industry. Even though many studies described the proof of purification for recombinant proteins such as EPO, they did not provide adequate information and data on whether their process is scalable and controllable (Table 1). In addition, validation is an integral part of a process; it involves the systematic study of systems, facilities, and processes with the target of determining whether they perform their intended functions properly and consistently as specified. A validated process should be capable of providing a high degree of assurance that uniform batches will be produced, and relevant products shall meet preset specifications. Validation itself cannot improve processes but confirms that the processes have been properly developed and are under control. Moreover, if the process is developed with appropriate CPPs then validation will confirm that the process is under “total control”. Biopharmaceutical industries can meet the QTPP and regulatory requirements by achieving a validated process embedded with well-characterized and defined CPPs [26,27]. Henceforth, it is extremely important to establish such a process for cost-effective EPO manufacturing which is scalable, controllable, and validated for satisfying QTPP, and will ultimately meet regulatory requirements. Here, we report such a robust and scalable EPO purification process embedded with defined CPPs and meeting QTPP at a lower risk and reduced cost.

## 2. Materials and Methods

### 2.1. Screening of Binding and Elution Conditions of EPO

Different chromatography resins, i.e., Capto blue affinity chromatography resin, Q Sepharose anion exchange chromatography resin, source reversed phase chromatography resin, and MacroCap SP cation exchange resin (GE Healthcare, Chicago, IL, USA) were selected, and different buffer conditions were applied as mentioned elsewhere in the manuscript to identify suitable condition for each resin to capture and elute the target product with reduced impurities. Approximately 100 µL of each resin was taken in 96-well plate during screening. Each resin was activated with equilibration buffer, and 1 mL of filtered (0.6 µm followed by 0.2 µm PES) sample was applied for interaction with resin. Samples were washed with equilibration buffer followed by elution with elution buffer. Buffer of AFC eluate was then exchanged by AEX equilibration buffer using 10 kDa NMWCO Vivaspin (Sartorius Stedim, Göttingen, Germany) column then applied in AEX resin; after washing with equilibration buffer, the products were eluted with elution buffer. AEX eluate was then applied to RPC resin, and the desired product was eluted after washing with equilibration buffer. RPC eluate was then applied to CEX resin; after washing the resin with equilibration buffer the products were eluted in elution buffer. The flow-through, wash, elution, and cleaning-in-place (CIP) samples of each chromatography resin were tested by dot blotting to identify the desired product. CIP was performed with 1 N NaOH solution for the AFC and AEX resins, whereas 0.5 N NaOH solution was used for RPC and CEX resins.

### 2.2. DoE for the Unit Processes of Purification Train

DesignExpert 13 software (Stat-Ease Inc., Minneapolis, MN, USA) was used for DoE. CPPs and optimum buffer conditions for AFC, AEX, RPC, and CEX chromatography were identified using 2-level factorial optimal (Custom) design. Data were analyzed using surface response methodology. Combinations of pH levels and NaCl concentrations for AFC (Appendix A) and AEX (Appendix A), pH and aquous:organic solvent for RPC (Appendix A), and pH and CV for CEX (Appendix A) were analyzed, respectively. The recommended chromatography conditions for each chromatography step were completed in static mode in 1 mL size in microcentrifuge tube. Recovery % of eluted samples of each step for all run conditions was analyzed by analytical size exclusion chromatography (SEC). Recovery data of target product satisfying QTPP for all chromatography steps were processed in DesignExpert 13 software. Optimum operating conditions and CPPs were identified from the surface response plot. All data points were validated using series of triplicate experiments.

### 2.3. Scale-Up and Adaptation of the Unit Purification Processes in Dynamic Mode

After finding the binding and elution conditions (design space) of product, the unit purification process steps were scaled up for the adaptation in dynamic mode from static mode. The purification process of EPO was optimized at 1× batch size (50 mL), which was compatible with small-scale column formats. The chromatography conditions were adapted through two consecutive small-scale (1×) batches and then validated with next three consecutive batches. After adaptation of process on small-scale, reproducibility of overall yield percentages among the batches were tested following first degree of statistical approach, and the acceptance criteria of those process parameters were set. The sample was sequentially filtered through 0.6 µm and 0.2 µm PES filter (Sartorius Stedim, Germany), and EPO was captured onto equilibrated AFC column (GE Healthcare, USA) in AKTA pure 25 system (GE Healthcare, USA) followed by washing and elution. FPLC systems and columns were sanitized before and after each purification step. FPLC results were evaluated through Unicorn 7.0 software (GE Healthcare, USA) for all chromatography steps. The AFC-eluates were buffer exchanged with 10 kDa Vivaspin protein concentrator (Sartorius Stedim, Germany) using Sorvall Lynx XTR centrifuge (Thermo Scientific, Waltham, MA, USA). The protein concentrator was sanitized before and after diafiltration. The retentate was loaded onto AEX column in AKTA pure 25 system. The AEX-eluate was then loaded onto RPC column in AKTA pure 25 system. The column was equilibrated and washed with acidic solution followed by elution in mixed gradient between wash buffer and organic solvent. The eluates from RPC were incubated for virus inactivation at low pH. The virus-inactivated samples were loaded onto CEX column in AKTA pure 25 system. The in-line pH, conductivity as well as sample in-line pH, conductivity, and quantity of buffers were considered as in-process check (IPC) points in AFC, AEX, RPC, and CEX steps. Residence times for chromatography were maintained at 6–10 min, and delta column pressure was maintained at below 6 bar. The samples were then passed through 0.22 µm filter followed by 20 nm ViroSart virus filter (Sartorius Stedim, Germany). The clarified samples were then reconstituted in formulation buffer. After passing through 0.45|0.2 micron Minisart sterilizing filter (Sartorius Stedim, Germany), samples were stored at 2–8 °C and analyzed.

### 2.4. Scale-Up and Validation of 10× (500 mL) and 100× (5000 mL) Batches

After adaptation of 1× batch in dynamic mode and optimization of operating conditions, the process was scaled up at 10× (500 mL) batch size by extrapolating CPPs for relevant unit operations. The sample was processed for AFC unit process and then subjected to AEX unit process after completion of tangential flow filtration (TFF). Sartocon Slice TFF cassette (Sartorius Stedim, Germany) and AKTA Flux 6 (GE Healthcare, USA) were used for buffer exchange in this unit process step. AEX eluates were subjected to RPC unit process followed by virus inactivation. Samples were then processed for CEX unit process and reconstituted in formulation buffer. All IPC parameters were applied in 10× batches. Three consecutive 10× batches were performed to validate the process, and samples were analyzed for QTPP. Similarly, all unit processes were further scaled up to 100× batch size (5000 mL) maintaining IPCs and CPPs, and samples were analyzed for QTPP. Appropriate quantity of resins, relevant columns, hardware, and buffers were used for successive scale-up of unit processes.

### 2.5. Analytical Approach for QTPP Confirmation

#### 2.5.1. Dot Blotting

PVDF transfer membrane (0.2 µm; Thermo Fisher Scientific, Waltham, MA, USA) was cut based on sample number and regenerated in methanol. The membrane was equilibrated in transfer buffer (pH 8.3). Subsequently, 10 µL of each sample was loaded on the membrane and allowed to dry. The reactivated membrane was blocked and treated anti-Epo polyclonal antibody (Thermo Fisher Scientific, USA). Goat anti-rabbit (H + L) IgG HRP conjugated (Thermo Fisher Scientific, USA) was used as secondary antibody. Novex^®^ ECL Chemiluminescent Substrate (Thermo Fisher Scientific, USA) for HRP was used to detect the signal, and the images were captured using Amersham Imager 600 RGB (GE Healthcare, USA).

#### 2.5.2. Particle Size Distribution

Samples were prepared in 0.22 micron filtered 1× PBS (pH 7.2), and after stabilization at 20 °C for 20 min, analyzed in disposable plastic cuvette using a Zetasizer Nano ZSP (Malvern Panalytical Ltd., Malvern, UK) where respective buffers used as dispersant. The equipment was switched on minimum of 30–60 min before the experiment to stabilize the system. The refractive index (RI), viscosity, and dielectric constant of dispersion buffer (1× PBS, pH 7.2 at 20 °C) were considered 1.33, 0.88 cPs, and 79, respectively.

#### 2.5.3. Chromatography for Determination of Assay and Impurities

The assay for EPO of the samples was determined using Vanquish UHPLC system (Thermo Fisher Scientific, USA). In short, the samples and reference product Eprex^®^ (20 µL of each) were applied in RPC column (Hypersil GOLD C8, 175 Å, 2.1 × 100 mm, 1.9 µm) (Thermo Fisher Scientific, USA) for analysis. The formulation buffer was considered as baseline reference. A gradient of mobile phase A (water with 0.1% FA) and mobile phase B (90% ACN in water with 0.1% FA) was used as carrier solvent at a flow rate of 0.3 mL/min. The impurity profiles of the samples were analyzed using SEC in reference to Eprex^®^. Analyses were performed in an Ultimate 3000 RSLC system (Thermo Fisher Scientific, USA) using 50 μL samples in a Biobasic SEC-300 column (300 mm × 7.8 mm, 5 µm) (Thermo Fisher Scientific, USA). Phosphate buffer (pH 7.4) was used as mobile phase at a flow rate of 1.0 mL/min. The column temperature was maintained at 70 °C, and run time was 25 min for both methods. The signals were detected at 280 nm, and chromatograms were recorded.

#### 2.5.4. In Vitro Functional Assay Using Cell Culture

The in vitro functional assay was performed using the TF-1 cell line (ATCC, Manassas, VA, USA; ATCC number: CRL-2003^TM^ and lot number: 64161542), which was originally derived from a patient diagnosed with erythroleukemia. TF-1 cells were maintained in RPMI 1640 medium (Gibco, Waltham, MA, USA) supplemented with 10% FBS (Thermo Fisher Scientific, USA), 1% PS (Thermo Fisher Scientific, USA), and 5 ng/mL human recombinant GM-CSF (Thermo Fisher Scientific, USA) at 37 °C and 5% carbon dioxide. Cells were washed twice in PBS to eliminate GM-CSF and then were seeded at a density of 10^5 cells/well in a 24-well tissue culture non-treated cell-culture plate. Cells were grown for 72 h in the presence or absence of originator (Eprex^®^) and EPO samples from each batch at indicated concentration. Cells were collected, centrifuged, re-suspended in 1 mL of PBS, and counted by Countess 2 automated cell counter (Thermo Fisher Scientific, USA).

#### 2.5.5. Sterility and Endotoxin Testing

Bacterial and fungal sterility were tested using direct inoculation technique [45]. Samples (1 mL) were inoculated in 10 mL of Tryptic Soya Broth (Sigma Aldrich, St. Louis, MO, USA) media for 14 days and absorbance were measured at 600 nm. Endotoxin in samples was tested using Pierce LAL Chromogenic Endotoxin Quantitation Kit (Thermo Fisher Scientific, USA) as per supplier’s instructions.

## 3. Results

### 3.1. Screening of Binding and Elution Conditions of EPO

After the screening of AFC resins, we found that 20 mM Tris-HCl, pH 7.4 was suitable for equilibration and washing for the selected matrix, and 1.5 M NaCl in wash buffer was appropriate for elution of the sample. After dialysis against Tris buffer at neutral pH, the conductance and pH were found ≥3.0 mS/cm and 7.0, respectively. For AEX chromatography, we observed that the similar buffer composition of AFC at a neutral pH was suitable for binding and elution of EPO. In the RPC chromatography process, 0.1% of TFA in WFI was found appropriate for resin equilibration and wash, and the product was eluted in 95% of acetonitrile. In the CEX chromatography process, 20 mM glycine, pH 2.0 was found suitable for the resin equilibration and wash. Sodium phosphate buffer with 150 mM NaCl, pH 7.20 was effective for eluting the EPO-satisfying QTPP. Representative samples of different process steps were identified with dot blot analysis (Figure 1).

### 3.2. Design of Experiments (DoE) of Purification Process

From the contour plot and 3D surface response plot, the pH range of media for the AFC unit process was extrapolated within 6.7–8.4 (control space), more preferably at 7.4 ± 0.2 (operation space).

The sodium chloride concentration range was extrapolated within 1400–1700 mM (control space), more preferably at 1500 ± 100 mM (operation space), which is equivalent to 120.4–146.2 mS/cm and 129 ± 10 mS/cm, respectively. The actual value and predicted value of recovery percentage were within close distribution to the regression line indicating that the pH and NaCl concentration both have a strong effect on AFC recovery (Figure 2). The model terms are significant where F-value was found to be 9.49 and the *p* value was 0.0015, suggesting that the model was well fitted. After analyzing eluted samples of all run conditions, we found less recovery with higher impurities at a pH lower than 6.7 with a 1500 mM (NaCl) carrier. In contrast, we have observed adequate recovery with higher impurities at a pH level above 8.4 with a 1700 mM (NaCl) carrier (Appendix A). The recovery range of the target product was found ≥80% with the least impurities in the control space and was considered an acceptable limit. For the AEX unit process, the buffer pH range was extrapolated within 6.5–7.7 (control space), more preferably at 7.0 ± 0.2 (operation space). NaCl concentration range was extrapolated within 260–380 mM (control space), more preferably at 300 ± 15 mM (operation space), which are equivalent to 22.5–32.70 mS/cm and 25.5 ± 1 mS/cm, respectively. The actual value and predicted value of recovery percentage were in a close distribution to the regression line indicating that the pH and NaCl concentration both have strong effects on AEX recovery within the operation range (Figure 3).

The model terms were significant where F-value was found to be 20.58 and the *p* value was <0.0001, suggesting that the model was well fitted. After analyzing eluted samples of all run conditions, we found less recovery with higher impurities at a pH lower than 6.5 with a 260 mM (NaCl) carrier. In contrast, adequate recovery with higher impurities were observed at a pH above 7.7 with a 380 mM (NaCl) carrier (Appendix A). The recovery range of the target product was found ≥40% with the least impurities in the control space and was considered an acceptable limit. For the RPC unit process, the pH range was extrapolated within 2.2–2.7 (control space), and more preferably at 2.4 ± 0.2 (operation space); whereas the acetonitrile concentration range was extrapolated within 45–67% (control space), and more preferably at 52 ± 5% (operation space). The actual value and predicted value of recovery percentage were found closer to the regression line within the operation range and suggested that the pH and concentration (%*v*/*v*) of acetonitrile both have a strong effect on RPC recovery (Figure 4).

The model terms were significant where F-value was found to be 6.20 and the *p* value was 0.0072, which suggested that the model was well fitted. After analyzing eluted samples of all run conditions, we found less recovery with higher impurities at a condition pH range lower than 2.2 and 45% acetonitrile. In contrast, adequate recovery with higher impurities were observed at a pH range above 2.7 with 67% acetonitrile (Appendix A). The recovery range of the target product was found ≥40% with the least impurity within the control space and was considered an acceptable limit. The pH range of buffer for CEX unit operation was extrapolated within 4.8–6.8 (control space), and more preferably at 5.8 ± 0.2 (operation space) though the plot showed a bimodal effect. The buffer volume was extrapolated within 2.5–4 CV (control space), more preferably at 3.5 ± 0.2 CV (operation space) to obtain the best outcome. The actual value and predicted value of recovery percentage were found closer to the regression line and indicate that the pH and buffer volume (CV) both have strong effects on CEX recovery (Figure 5). The model terms were significant where F-value was found 176.36 with a *p* value < 0.0001, which suggested that the model was well fitted. After analyzing eluted samples of all run conditions, we found less recovery with impurities at a pH lower than 4.8 with 2.5 CV of buffer volume. In contrast, adequate recovery with higher impurities were observed at a pH above 6.8 with 4 CV of buffer volume (Appendix A). The recovery range of the target product was found ≥80% with the least impurities within the control space and was considered an acceptable limit.

### 3.3. Adaptation of Unit Process in Dynamic Mode, and Scale-Up

Five consecutive batches were conducted to adapt the process in dynamic mode using chromatography column format and machine-controlled flow conditions, and the acceptance criteria were justified for each unit process step that was identified from DoE runs. The quantity of the starting materials of these five batches was 45–55 mg, the volume was approximately 50 mL and the concentration of EPO was 0.9–1.1 mg/mL (Appendix A). In the case of the AFC unit process, it was observed that the pH range of the elution buffer and eluted sample were 7.43 ± 0.12 and 7.35 ± 0.14, respectively. The conductivity range (mS/cm) of the elution buffer and sample were 123.42 ± 1.61 and 123.78 ± 1.22, respectively. The average eluted protein quantity and yield % were found 41.62 ± 2.38 mg and 81.11 ± 2.66 mg, respectively. The pH and conductance were considered as CPPs and % of yield was considered one of the CQAs for this step (Appendix A). The pH range of the retentate sample, conductance (mS/cm), quantity (mg), and yield % were 7.35 ± 0.15, 2.53 ± 0.29, 41.09 ± 2.27, and 98.73 ± 0.31, respectively, where conductance of the retentate sample was considered as CPP (Appendix A). Since, the retentate sample pH, conductivity range, and yield percent were not evaluated in DoE (in static mode), the average value and standard deviation of the parameters of these five batches were considered as acceptance limits. In the AEX unit process, elution buffer and eluted sample pH were found 7.05 ± 0.14 and 6.97 ± 0.11, respectively, whereas, the conductivity (mS/cm) range was found 123.11 ± 1.33 and 24.09 ± 0.69, respectively. The average quantity of eluted protein was 16.86 ± 1.24 mg with a yield % of 41.01 ± 1.03. The pH of the buffer and the conductance of the eluted sample were considered as CPP, whereas % yield was considered as one of the CQAs for this unit process (Appendix A). The pH, conductivity, and yield percent were within the acceptable limit which was identified through DoE. In the RPC unit process, the pH range of the elution buffer and eluted sample were observed 2.43 ± 0.09 and 2.17 ± 0.11, respectively. The acetonitrile % of the eluted sample was found 50.80 ± 1.92. The average eluted protein quantity and yield % were 8.68 ± 0.57 mg and 51.51 ± 1.72, respectively. The pH of the buffer and % (*v*/*v*) of acetonitrile were considered CPPs, and yield % was considered as one of the CQAs (Appendix A). The pH of the elution buffer, % (*v*/*v*) of acetonitrile, and yield % were within acceptable limits, which were identified as DoE output. After completion of RPC eluted sample was kept for 120 min to inactivate the virus particle and then proceed to CEX. For the CEX unit process, it was found that the pH range of the elution buffer and the eluted sample was 7.07 ± 0.20 and 5.81 ± 0.11, respectively. A 3.5 CV of elution buffer was found sufficient to elute desired product where average eluted protein quantity and yield % were 7.47 ± 1.02 mg and 85.86 ± 7.42, respectively. The pH of the elution buffer and buffer volume was considered as CPPs and % of yield was considered as one of the CQAs for this step (Appendix A). The pH and the buffer volume (CV) of the elution buffer and yield % were within acceptable limits, which were identified as DoE output. In the virus filtration (VF) step, the pH, conductance (mS/cm), quantity (mg), and yield % were 7.12 ± 0.12, 12.54 ± 0.43, 7.38 ± 0.97 and 98.80 ± 0.65, respectively. The pH of the filtrate sample and time were considered as CPPs, whereas the % of yield was considered one of the CQAs (Appendix A). Since the % yield was not evaluated in DoE (in static mode), the average value and standard deviation of the parameters of these five batches were considered acceptance limits. After completion of the sterile filtration (SF) step, it was observed that the pH and conductance of the sample and yield % were similar to the previous step (Appendix A). The overall yield % for the whole process was calculated to be 14.14 ± 0.87% (Appendix A). Since the overall % yield was not evaluated in DoE (static mode), the average value and standard deviation of the yield % of these five batches were considered acceptance limits. For all the dynamic unit processes, the CPPs were controlled and monitored using an inline automated system.

### 3.4. Validation of Unit Processes for 10× (500 mL) Batch

After completion of the purification process optimization, batch size was scaled up to 10× (500 mL), and three consecutive batches (Batch No. 06, 07, and 08) were conducted to validate the purification process. The overall yield percentage of batch no. 06, 07, and 08 were found 13.97%, 14.07%, and 13.90%, respectively, which were within the acceptance range (14.14 ± 1.0%) of yield percentage (Appendix A). The recovery percentage data for individual unit process steps were also within the limit for these three batches (Appendix A). IPC parameters of chromatography steps were monitored and controlled by the inline FPLC system during the batch runs.

### 3.5. Validation of 100× (5000 mL)

The batch size was further scaled up to 100× (5000 mL), and three consecutive batches (Batch No. 09, 10, and 11) were conducted to validate the scaled-up process. The overall yield percentage of batch no. 09, 10, and 11 were found 13.30, 14.62, and 13.31%, respectively, which were within the acceptance range (14.14 ± 1.0%) of the yield (Appendix A). The recovery percentage data for individual unit process steps were also within the limit for these three batches (Appendix A). All IPC parameters of chromatography steps were monitored by an automated inline FPLC system for these batches. The representative chromatogram is shown in Figure 6A–D.

There were no significant differences observed for the qualitative and quantitative parameters of each unit processes of 1×, 10×, and 100× validation batches (Figure 7A, *p* > 0.05). We also found that the variation between three batches for each batch size (1×, 10×, and 100×) is insignificant (Figure 7B, *p* > 0.05). The yield percentage of the three scaled-up batches for both scales (10× and 100×) have no significant variation either (Figure 7C, *p* > 0.05). The trend analysis for all the batch data for yield % were within the control limit (acceptance range 13–15%) (Figure 7D).

### 3.6. Analysis for QTPP

After confirmation of the validation of 1× (50 mL), 10× (500 mL), and 100× (500 mL) batch size, relevant samples were analyzed as per specification [45] and found all parameters of the specification were in accordance with the relevant acceptance limits (Appendix A). The particle size distribution was compared between reference product Eprex^®^ and different batches of GBPD002, where representative results were shown in Appendix A. The representative result for the identification test, impurities profile, and biofunctionality analysis between reference product Eprex^®^ and GBPD002 were shown in Appendix A. RP-UHPLC analysis data provided a clear understanding that a similar quantity of EPO was harvested and was present in each batch (Appendix A). The SEC-UHPLC result did not show any significant differences between the GBPD002 harvested from different batches and with the reference product, suggesting that there are no unwanted species present in any of the EPO preparations (Appendix A). Comparing the buffer chromatograms, the secondary peaks for both RPC and SEC can be attributed to the buffer and carrier system. All these analyses collectively confirmed the absence of any macromolecular entity that can be generated from process chemistry or product aggregates. The TF-1 cell proliferation data clearly revealed that the materials from different batches and references responded similarly. Compared with the mock-controlled dishes, which have reduced to half by the number from its original seeding population (10^5/well), all samples assisted the growth of the TF-1 cell population to three times the originally seeded cell numbers (Appendix A). This data has clearly established that the EPO preparations are similarly active to the reference product. Very close responses for the biofunctionality among the EPO preparations from three different batch sizes (1×, 10×, and 100×) suggested indifferences among these scaled-up EPO preparations (Appendix A, *p* > 0.05).

## 4. Discussion

We have taken a systematic pathway (Appendix A) to address the challenges associated with the unit processes for the purification of EPO and transform the process train to higher batch sizes with the objective of meeting regulatory requirements. As per WHO guidelines for biologic/biosimilar drug evaluation, the quality attributes of biologics can be furnished into three categories, viz., (i) very high critical, (ii) high critical, and (iii) low critical [46]. Amino acid sequence, glycan analysis, biological activity, and immunochemical identity are considered very high critical quality attributes. Higher order structure, isoform distribution, insoluble aggregates, high molecular weight aggregates, protein content, host cell proteins, host cell DNA, and receptor binding are considered highly critical quality attributes. Deamination, oxidation, and truncation are considered low critical quality attributes. The amino acid is produced based on the stably integrated relevant DNA of the qualified master cell bank (MCB), and we have found the expressed rhEPO indeed contains all 165 amino acids by LC-MS/MS [45]. The MCB has been characterized for 50 consecutive passages over 160 days [45]. Therefore, it is highly unlikely that the protein sequence is compromised in the current study, where we have used a 2-week culture of qualified seeds from the qualified working cell bank (WCB).

The glycosylation profile has been found important for protein folding that affects protein stability and receptor interaction. We did not consider glycoform as a QTPP parameter for this study due to the following reasons. Firstly, though glycoforms analysis are important and may be required for batch release but in practical aspect it is a non-critical parameter for our study as described follows. It has been found that several different EPO-glycoforms were present in the circulating systems of humans [47], and the sugar profiles of human serum EPO are significantly different from the profiles of rhEPO [48]. Despite having differences in sugar profile, there were no significant differences reported for in vivo biofunctionality for several preparations (12 different sources) of rhEPO [49]. Several clinical studies also did not reveal any significant differences between pharmacokinetic and pharmacodynamic properties for different rhEPO [50,51,52,53,54,55,56]. Secondly, confident glycoform analysis is a very critical process, and multiple parallel methods are preferred for obtaining confident PTM profiles using mass spectrometry analysis [56,57,58]. Further, it has been suggested that the separation of recombinant EPO into pure glycoforms is not feasible even with the most advanced methods due to the heterogeneity of the three N-glycans [59]. Thirdly, the objective of the study was not the thorough qualification of a candidate EPO in respect of biosimilarity but rather to harvest EPO through a well-characterized and scalable downstream process and use some suitable analytical methods for analyzing the harvest which may conform with the preset specification of EPO.

We have considered aggregation analysis for the QTPP profile of EPO due to the fact that aggregation mainly occurred during the downstream processing particularly, during and on the formulation. EPO aggregation has been reported that is critically connected to the immunochemistry of rhEPO preparation; it was specifically responsible for the development of EPO antibody-mediated deadly disease condition called pure red cell aplasia (PRCA) [60,61,62]. Due to the criticality of the parameter, we have employed three orthologous analysis procedures, viz dynamic light scattering, and SEC-HPLC for detecting any high molecular aggregates in our final products, and all analyses confirmed that our EPO preparations from all three scales did not contain any such macromolecular entity.

Quantity and relevant impurities have been considered highly critical quality attributes, and therefore, we considered these parameters as the components of QTPP in our study. We have used orthologous UHPLC protocols in two different methods (RPC and SEC) for analyzing these QTPP parameters. All samples from three different scale size (1×, 10×, and 100×) batches generated highly similar chromatograms like those of the reference product. The results suggested indifferences between experimental EPO preparations and references in terms of two critical QTPP parameters, viz., quantity and purity. Since EPO biofunctionality is a concrete proof of functional significance, we, therefore, considered the cell-based biofunctional assay as another parameter for QTPP and found similar results for all batches and reference products. The QTPP profile for all validation and scale-up (1×, 10×, and 100×) batches satisfied the regulatory acceptance limit of relevant analyses, and thereby proved that the QbD approach used to develop unit processes is effective. Insignificant levels or minor variations in different data points were not affecting the final product quality. In fact, such types of minor variations are not uncommon for US FDA and EMA-approved biosimilars [63]. A recent study compared the quality and batch-to-batch variability of marketed rhEPO reference medicines, Eprex^®^/Erypo^®^ (Janssen-Cilag, High Wycombe, UK) and NeoRecormon^®^ (epoetin beta; Roche Registration Limited, Welwyn Garden City, UK), and two biosimilars, Binocrit^®^ (Sandoz GmbH, Kundl, Austria) and Retacrit^®^ (Hospira UK Limited, Maidenhead, UK), and found batch–batch minor variability for experimental products [64]. Microbial contamination and pyrogen are another two QTPP profiles we have included in our study. These contaminants in parenteral drug preparation are life-threatening and mainly enter into the product through the process therefore these parameters are indispensable from QTPP. Relevant results revealed that all EPO preparations were clear of such contaminants and thereby satisfy relevant QTPP.

Several studies have reported the purification of rhEPO; a snapshot of comparative process analysis has been shown in Table 1. The pros and cons of individual processes have been identified and summarized therein. Relevant studies have reported the purification processes of rhEPO with different combinations of multiple unit processes and conditions but none of them has systematically developed their unit processes that may qualify for a regulatory framework. Neither the CPPs were identified nor the operating space for any relevant unit processes were characterized in these studies, thereby severely limiting the scale-up and scale-down opportunities. Several processes are SEC dependent [28,29,30,31,32,33,34], and SEC is not economic in biopharmaceutical processes since it requires a huge amount of resin and larger columns to handle bulk samples. It also increases process time since it needs to run at a slow flow rate because of increasing delta column pressure. Some processes placed the SEC unit process after the RPC unit process (where the product is in low pH) [28,29], which is detrimental to the product since EPO degrades fast in low pH. We have avoided expensive and highly critical SEC processes and aligned the other processes in the process train to exploit the physicochemical properties of the output of an antecedent unit process as the input for the subsequent unit process, which made the process cost-effective and easily scalable. Many processes did not mention virus inactivation and specific step for virus reduction as well as the elimination of microbial and pyrogenic contaminants [30,31,32,33,34], which is a big concern for the regulatory framework to ensure QTPP and patient safety. Our process includes all relevant steps that make the end product safer for patients. Several processes were not aligned with regulatory guidelines since the final product was not in formulated forms and they did not include impurity profile and viral and microbial decontamination processes [35,36,37,38,39,40,41,42,43]. In 2017, Bandi et al. claimed 98% purity for their process but this process has lots of drawbacks. It is associated with at least three buffer exchange steps (in between AEC and CEX, CEX and AFC, and AFC and AEX), totaling ten steps including SEC steps, which is a very expensive and lengthy process though this process is not completed up to the formulation step [44].

Our process includes eight steps to obtain ready-to-fill EPO preparations from the filtered media upstream satisfying QTPP in line with the regulatory framework. Identification of CPPs by DoE in our process made it efficiently convenient to scale-up and validate. After completion of validation, variations between the individual process steps for 1×, 10×, and 100× batches were insignificant (*p* > 0.05), as well as the variations within the batches were also insignificant (*p* > 0.05), which suggested that data were reproduced for 1×, 10×, and 100× batches. Collectively, our study provided a scalable and controllable downstream process for the purification of drug-quality rhEPO to satisfy regulatory requirements at a competitive cost.

Moreover, we evaluated the product through a clinical trial where we obtained satisfactory results where the PK, PD, and safety profiles of the test (GBPD002) epoetin alfa were found similar to the comparator (Eprex^®^) [65] and the ClinicalTrials.gov Identifier number is NCT05585658, National Library of Medicine, National Institute of Health (NIH), USA [66]. It has been reported that the PK, PD, and safety profiles are indifferent for males and females [67]. Therefore, the clinical data observed for the product GBPD002 can be considered sex independent.

## 5. Conclusions

Here, we presented a systematic approach to developing a well-characterized and scalable purification process of EPO preparation for pharmaceutical use. The operation boundaries for each unit process were validated in two steps (10× and 100×) of the scale-up process resulting in a 100× batch size, which provides a tremendous opportunity for seamless integration of the process train at the industrial scale. As a result, EPO can be manufactured in a faster time and cost-effective manner. The process development and validation pathway presented here can be applied to many other life-saving pharmaceutical products and thereby will facilitate the relevant downstream process management and promote the availability of cost-effective products to the global community.

## Figures and Tables

**Figure 1 pharmaceutics-15-02087-f001:**
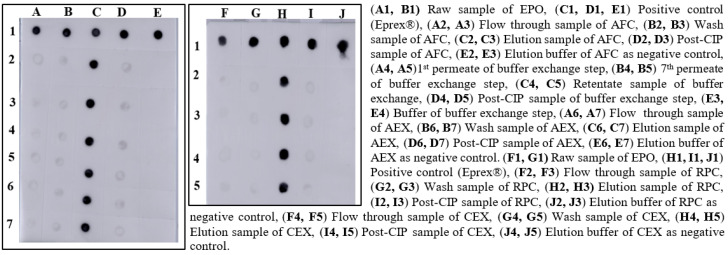
Screening of binding and elution conditions of EPO using different resins.

**Figure 2 pharmaceutics-15-02087-f002:**
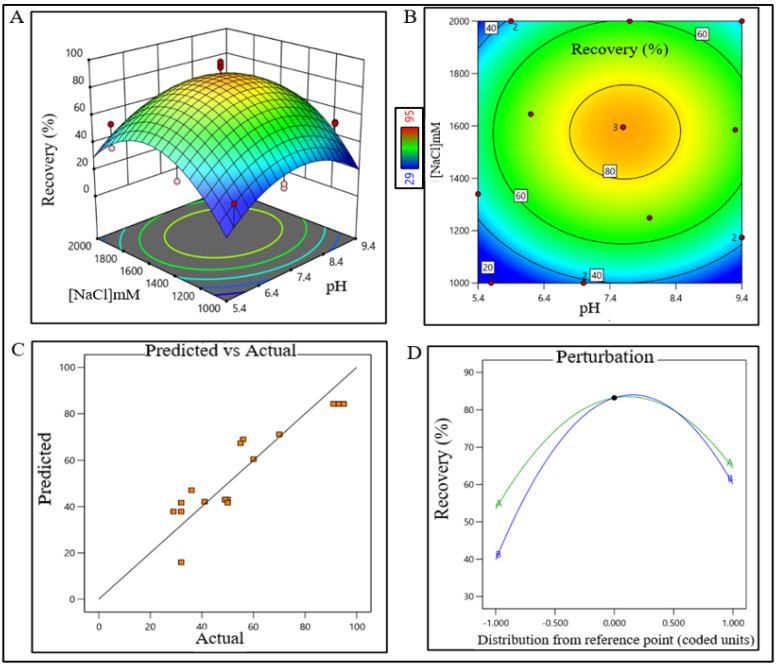
DoE surface response plot of AFC process step in static mode. (**A**) Three-dimensional surface response plot of elution condition, (**B**) contour plot of elution condition, (**C**) actual vs. predicted response of recovery %, and (**D**) perturbation of recovery % against two factors pH (line A) and [NaCl] (line B).

**Figure 3 pharmaceutics-15-02087-f003:**
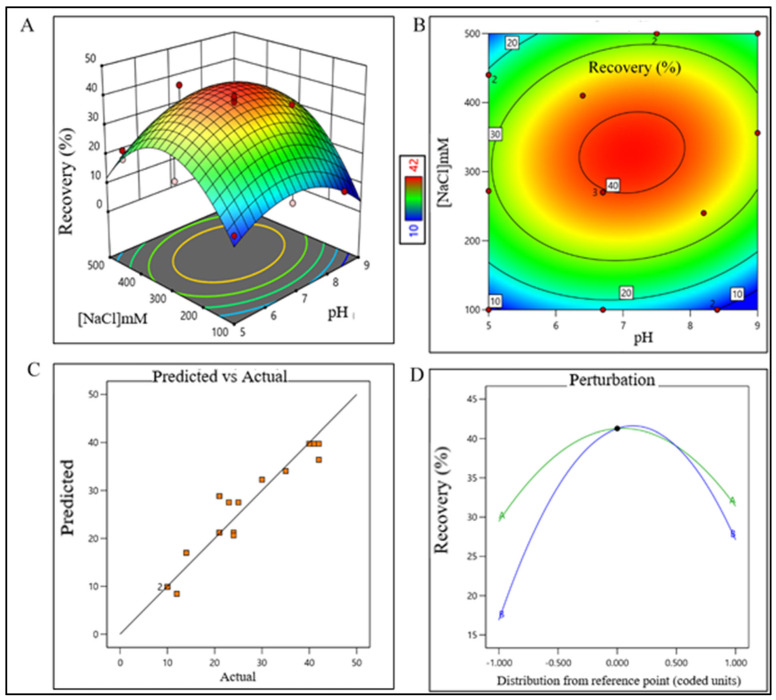
DoE surface response plot of AEX process step in static mode. (**A**) Three-dimensional surface response plot of elution condition, (**B**) contour plot of elution condition, (**C**) actual vs. predicted response of recovery %, and (**D**) perturbation of recovery % against two factors pH (line A) and [NaCl] (line B).

**Figure 4 pharmaceutics-15-02087-f004:**
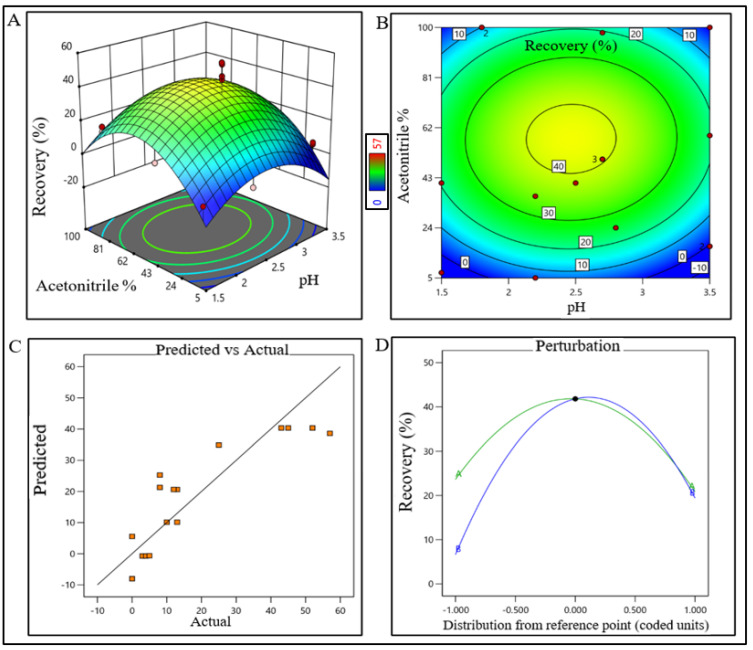
DoE response plot of RPC process in static mode. (**A**) Three-dimensional surface response plot of elution condition, (**B**) contour plot of elution condition, (**C**) actual vs. predicted response of recovery %, and (**D**) perturbation of recovery % against two factors pH (line A) and acetonitrile% (line B).

**Figure 5 pharmaceutics-15-02087-f005:**
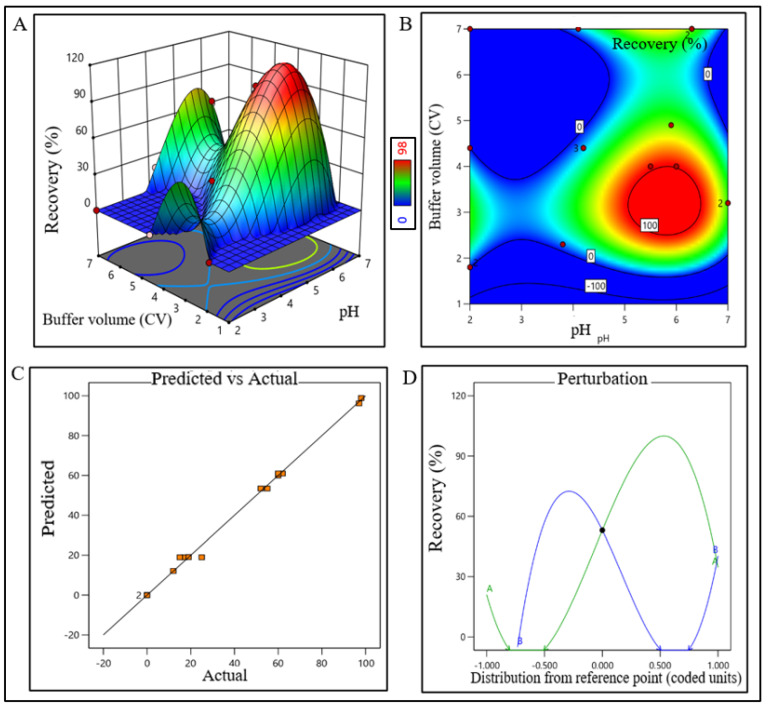
DoE response plot of CEX process in static mode. (**A**) Three-dimensional surface response plot of elution condition, (**B**) contour plot of elution condition, (**C**) actual vs. predicted response of recovery %, and (**D**) perturbation of recovery % against two factors pH (line A) and buffer volume (CV) (line B).

**Figure 6 pharmaceutics-15-02087-f006:**
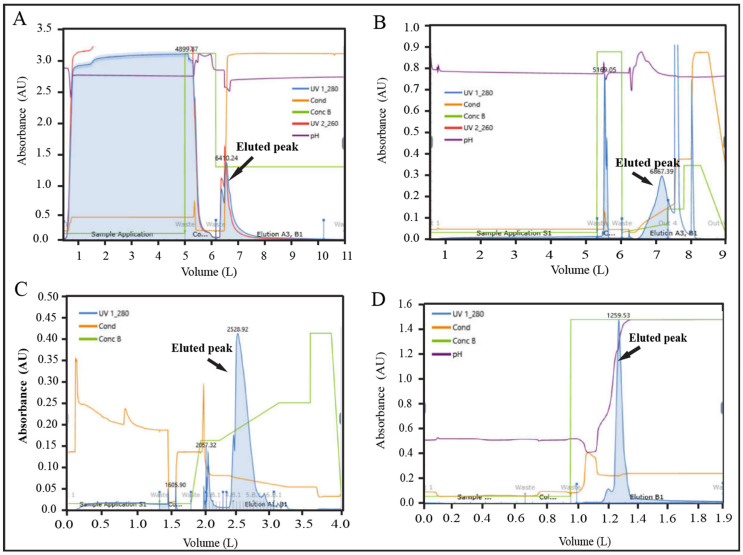
Representative FPLC chromatograms of validation batch (100×). (**A**) AFC chromatogram, (**B**) AEX chromatogram (**C**) RPC chromatogram, and (**D**) CEX chromatogram.

**Figure 7 pharmaceutics-15-02087-f007:**
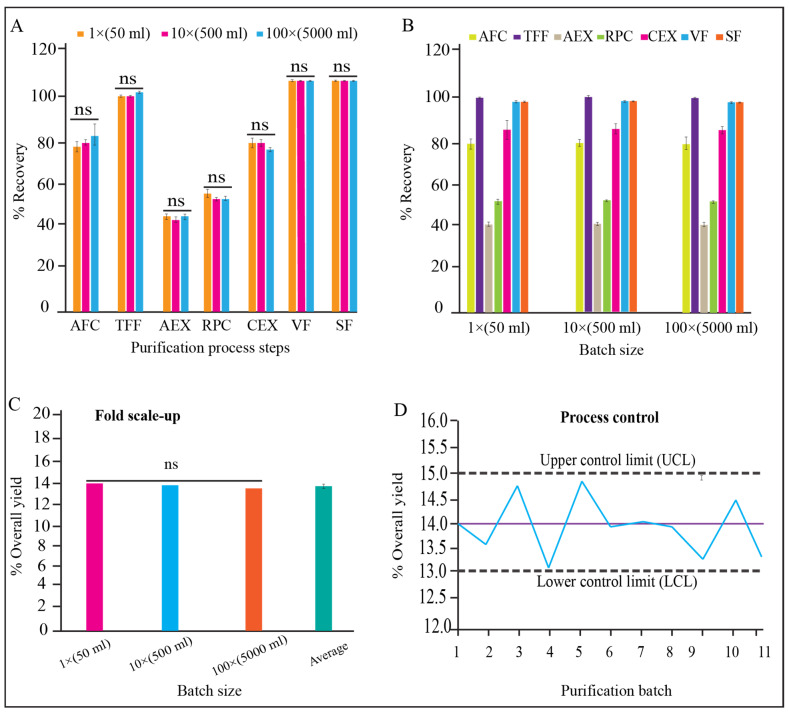
Process monitoring and qualification. (**A**) Recovery % of each step in dynamic mode, and reproducibility between process steps of 1×, 10×, and 100× batch size, (**B**) recovery % of each process step and reproducibility within batches of 1×, 10×, and 100× batch size, (**C**) reproducibility of yield between 1×, 10×, and 100× batch, (**D**) controllability of purification process for 1× (batch 1–5), 10× (batch 6–8), and 100× (batch 9–11) batch sizes, shown in “x” axis.

**Table 1 pharmaceutics-15-02087-t001:** Different purification processes for EPO with their pros and cons.

References	Process Steps	Process Performance	Limitations
Lai and Strickland et al. (1985) [28]	AEC → RPC →SEC	Not mentioned.	Incomplete and uncharacterized process, SEC is not economic or suitable for scale-up
Beck and Withy (1988) [29]	CEC → RPC → SEC	Total yield (%): 60 → 50 → 41	Incomplete and uncharacterized process. Need to adjust sample pH below 4 resulting in precipitation or degradation, SEC is not economic or suitable for scale-up
Sugaya et al. (1997) [30]	AFC → AFC → SEC	Total yield (%): 79 → 54 → 40	Incomplete and uncharacterized process, SEC is not economic or suitable for scale-up
Zanette et al. (2003) [31]	AFC → AEC → SEC	Total yield (%): 76 → 60 → 30	Incomplete and uncharacterized process. Need to adjust sample pH below 4, resulting precipitation or degradation, SEC is not economic or suitable for scale-up, impurities were not removed based on hydrophobicity
Hu et al. (2004) [32]	AEC → HIC → SEC	Total yield (%): 51 → 46 → 43	AFC is more efficient to enrich glycosylated protein compared to AEC, SEC is not economic or suitable for scale-up, AEC is more suitable to separate charge variant compared to HIC
Carcagno et al. (2006) [33]	HIC → AEC → CEC → SEC	Total yield (%): 70 →56 → 40 → 30	Need to add high molar salt at first step, SEC is not economic or suitable for scale-up, need extra process step to complete the process
Goletz and Stöckl (2012) [34]	RPC → AFC → SEC → AEC	Final host cell protein content less than 0.01%	AFC is more efficient to enrich glycosylated protein compared to RPC, SEC is not economic or suitable for scale-up and not suitable after AFC,
Wojchowski et al. (1987) [35]	HIC→ AFC	Total yield (%): 100 → 55	Incomplete and uncharacterized process, no suitability in production scale
Ghanem et al. (1994) [36]	AFC → AEC	Total yield (%): 73 → 53	Incomplete and uncharacterized process, no suitability in production scale
Hsu and Chang (2002) [37]	AFC → SEC	in vitro activity: 240 kU/mg	Incomplete and uncharacterized process, not suitable for scale-up, yield was not claimed
Surabattula et al. (2011) [38]	AFC → AEC	Final total yield: 42%	Incomplete and uncharacterized process, not suitable for scale-up
Broudy et al. (1988) [39]	AFC → AEC → RPC	Total yield (%): 65 → 49 → 35	After RPC, acetonitrile removal is difficult to formulate the sample, needs other steps which are not performed and characterized
Quelle et al. (1989) [40]	AEC → RPC → AFC	Total yield (%): 95 → 85 → 80	AFC is more efficient to capture glycosylated protein, difficult to formulate the sample after AFC, need other steps which are not performed and characterized
Merrifield (1990) [41]	AFC → AFC → AEC	Total yield: 80% (in the first step)	Need to add extra TFF process before each chromatography step, difficult to formulate the sample after AFC, need other steps which are not performed and characterized
Burg et al. (2002) [42]	AFC → HIC → HAC → RPC → AEC	Final total yield: 25%	Need other steps to formulate the sample which are not performed and characterized, after RPC, elute is not suitable to proceed for AEC
Koh et al. (2014) [43]	AEC → HAC → AEC	Total yield (%): 60.1 → 54.2 → 18.1	AFC is more suitable for enrichment of glycosylated protein, needs other steps which are not performed and characterized
Bandi et al. (2017) [44]	AEC *→* MMC *→* AEC *→* CEC *→* AFC *→* AEC *→* SEC	Purity (%): ≥98%	Need at least three TFF process steps and other steps to formulate the sample which are not characterized, too many process steps increase process cost, SEC is not economic or suitable for scale-up, specific yield was not claimed

## Data Availability

The data that support the findings of this study are available within the article and its Appendix A or are available from the corresponding author upon reasonable request.

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
