# Peer review of "Satisfying QTPP of Erythropoietin Biosimilar by QbD through DoE-Derived Downstream Process Engineering"

_pharmaceutics, 2023, doi:10.3390/pharmaceutics15082087_

Round 1

Reviewer 1 Report

This is a very interesting submission that presents a complete and well studied process for purification of EPO (and biologics in general) with reduced cost, following QbD principles. There is no major objection from my side for the experimental part, however, the text requires major revision in terms of presentation of results and towards decrease of its size.

Comments:

-Table 1 is mentioned in Introduction and appears many pages later. I believe it can be moved earlier in Introduction, even though there is relevant text in Discussion. By the way, the comparison with other methods is very important.

-On the contrary, there is no need to mention Scheme 1 in Introduction.

-Line 119: leave space between number and unit. The same applies to other lines, as well.

-Line 135: use 'methodology' instead of 'plot'

-Use a Table to include statistics of models for all 4 designs. The type should be written (i.e. quadratic), p value for model, p value for lack of fit and R^2, adj. R^2 and pred. R^2. This will decrease the size of text.

-Furthermore, all these graphs from Fig. 2-5 are not needed. It can be combined all in 1 figure by just using (A) plot of each Figure.

Fig. 6 should be split into two separate.

Fig. 7 can be moved to Supplementary material or replaced with description in text (or both)

Fig. 8 should be provided again. It should be more clear with less details. At least 2 out of 4 graphs should be removed.

Fig. 1 in Supplementary Material could be transferred in main text following results/discussion of DoEs.

Author Response

Comments and Suggestions for Authors:

This is a very interesting submission that presents a complete and well studied process for purification of EPO (and biologics in general) with reduced cost, following QbD principles. There is no major objection from my side for the experimental part, however, the text requires major revision in terms of presentation of results and towards decrease of its size.

Point 1: Table 1 is mentioned in Introduction and appears many pages later. I believe it can be moved earlier in Introduction, even though there is relevant text in Discussion. By the way, the comparison with other methods is very important.

Response 1: We highly appreciate your suggestion. It does not affect the scientific value of the manuscript either it is placed in Introduction or Discussion section. However, at its current place, Table 1 is easier to understand and convenient to follow the relevant discussion for readers; therefore, we placed it in Discussion section.

Point 2: On the contrary, there is no need to mention Scheme 1 in Introduction.

Response 2: Thank you very much for your suggestion. We agree to your suggestion and placed the Scheme 1 from Result section to Supplementary section as Supplementary scheme 1, as well as removed relevant text (Line 250 – 253). We think that it is very beneficial for general readers to understand the way of process development strategy along with process flow, and therefore, provided it in Supplementary section.

Point 3: Line 119: leave space between number and unit. The same applies to other lines, as well.

Response 3: Thank you very much for pointing out this typo. We corrected the space between number and unit as per your suggestions in manuscript.

Point 4: Line 135: use 'methodology' instead of 'plot'

Response 4: Thank you very much. We replaced the word ‘plot’ with ‘methodology’.

Point 5: Use a Table to include statistics of models for all 4 designs. The type should be written (i.e. quadratic), p value for model, p value for lack of fit and R^2, adj. R^2 and pred. R^2. This will decrease the size of text.

Response 5: Thank you very much for your valuable suggestion. We included the experiment model name, p value for model, p value for lack of fit and R^2, adj. R^2 and pred. R^2 in Supplementary table 1 – 4 in Supplementary section, respectively.

Point 6: Furthermore, all these graphs from Fig. 2-5 are not needed. It can be combined all in 1 figure by just using (A) plot of each Figure.

Response 6: Thank you for expert opinion. In Fig. 2 – 5, we showed 4 presentations as follows:

  1. A) 3D surface response plot, which represents data distribution with amplitude of impacts of factors
  2. B) Contour plot, which represents the central-tendency of data with the operation range of factors
  3. C) Actual vs. predicted response, which represents the experimental outcome with the predicted data and also represents how well the suggested conditions throughout the model is fitted with the experimental value
  4. D) Perturbation, which represents the relationship and effects of all the factors at a particular point in the design space

We aggree that the 3D surface response plot (A plot), as you have suggested, is sufficient to understand relevant sections for DoE experts. However, since these 4 figures represents 4 different types of significances (as mentioned above), therefore we have provided these presentations for easy understanding and interpretation of DoE model and results for general readers.

Point 7: Fig. 6 should be split into two separate.

Response 7: Thank you for your suggestion and relevant figure is divided into two separate figures (Figure 6 and Figure 7).

Point 8: Fig. 7 can be moved to Supplementary material or replaced with description in text (or both)

Response 8: We moved Fig. 7 in Supplementary material as Supplementary figure 2.

Point 9: Fig. 8 should be provided again. It should be more clear with less details. At least 2 out of 4 graphs should be removed.

Response 9: Thank you for your suggestion. We moved Figure 8 to Supplementary material as Supplementary figure 3 - 5. In addition, we provided clearer figures. We could not remove those data due to the regulatory concerns what mentioned in Line 549 and 555 in Discussion part.

Point 10: Fig. 1 in Supplementary Material could be transferred in main text following results/discussion of DoEs.

Response 10: We appreciate your suggestion. However, since relevant material is representative supporting analysis data of DoE output therefore, we placed it in Supplementaty material. We also believe that the relevant section at its current place does not dilute the scientific value of the article.

Reviewer 2 Report

Whereas the work seems to be carefully done, some few points need attention before publication. Please find bellow some suggestions to improve the quality of the manuscript.

1)    The authors should use rm2 metrics for validation. See J Comput Chem 34, 2013, 1071-1082 and Journal of Chemistry, v. 2016, p. 1-12, 2016 (http://dx.doi.org/10.1155/2016/9198582). All suggested references should be included in the paper as well.

Author Response

Responses to Reviewer 2 comments

Comments and Suggestions for Authors:

Point 1: Whereas the work seems to be carefully done, some few points need attention before publication. Please find bellow some suggestions to improve the quality of the manuscript.

1)    The authors should use rm2 metrics for validation. See J Comput Chem 34, 2013, 1071-1082 and Journal of Chemistry, v. 2016, p. 1-12, 2016 (http://dx.doi.org/10.1155/2016/9198582). All suggested references should be included in the paper as well.

Response 1: Thank you for your suggestion. We used Design Expert 13 software to perform DoE and this software utilizes rm2 metrics logics for model validation which is relevant to your mentioned references. Furthermore, we added Model name, p value for model, p value for lack of fit, R2, adjusted R2 and predicted R2 values in relevant Supplementary table 1 – 4.

Reviewer 3 Report

According to this manuscript, some minor points should be modified.

1. Title and keywords: the abbreviation should be presented the full name.

2. 2.5.5: TF-1 cells were from human that need to provide IRB approval certification (number).

3. Please correct mL or ml through the full manuscript.

4. line 238: 10^5.

5. Figs. 6-8: to replace the more clear Figures.

Author Response

Responses to Reviewer 3 comments

Comments and Suggestions for Authors:

According to this manuscript, some minor points should be modified.

Point 1:  Title and keywords: the abbreviation should be presented the full name.

Response 1: Thank you for your valuable comments. The abbreviated words mentioned in ‘Title’ are commonly used in regulatory framework, pharma industry, and easily understable to the relevant scientific and professional communities. Relevant people are more comfortable and fluent with these abbreviated forms instead of their full forms. However, for general readers we have mentioned the full forms in Abstract and Keyword (newly added in revised manuscript) sections.

Point 2: 2.5.5: TF-1 cells were from human that need to provide IRB approval certification (number).

Response 2: Thank you for your expert opinion. We purchased TF-1 cell line from ATCC in USA which was originally derived from a patient diagnosed with erythroleu-kemia and ATCC cell line reference number: CRL-2003TM and lot number: 64161542. We added source and origin of TF-1 cells in manuscript in ‘Materials and Methods’ section keeping track changes (Line 233 – 234).

The above mentioned human cell line was approved by Internal Ethical Clearance Board (IECB), Globe Biotech Limited, Bangladesh for research purposes (Protocol number# GB/EC/18/001 and approval date: February 15, 2018). The protocol number is also included in ‘Institutional Review Board Statement’ section (Line 645 – 647).

Point 3: Please correct mL or ml through the full manuscript.

Response 3: Thank you very much for pointing out relevant errors. We corrected relevant texts throughout the manuscript.

Point 4: line 238: 10^5.

Response 4: Thank you very much. We corrected relevant text.

Point 5: Figs. 6-8: to replace the more clear Figures.

Response 5: Thank you for your suggestion. We provided more clear figures.

Additionally, we have updated relevant figures as per suggestions of other reviewers: (a) Figure 6 is divided into two separate figures (Figure 6 and Figure 7); (b) Figure 7 and Figure 8 are moved to Supplementary sections where Figure 8 is divided into 3 Suplmentary figures (Suplmentary figure 3 – 5).

Reviewer 4 Report

The article entitled "Satisfying QTPP of Erythropoietin Biosimilar by QbD through 2 DoE-derived Downstream Process Engineering" is interesting.

The study thereby established a process of EPO biosimilar satisfying QTPP, the technological scheme presented here can speed up the production of  EPO and also many other life-saving biologics.

The discussion could be improved, by adding a small section about the impact of the use of this synthetic EPO in the population, the difference between the two sex (if it is available), and the difference and the health benefits for the patients.

Author Response

Responses to Reviewer 4 comments

Comments and Suggestions

The article entitled "Satisfying QTPP of Erythropoietin Biosimilar by QbD through 2 DoE-derived Downstream Process Engineering" is interesting.

Point 1: The study thereby established a process of EPO biosimilar satisfying QTPP, the technological scheme presented here can speed up the production of  EPO and also many other life-saving biologics.

The discussion could be improved, by adding a small section about the impact of the use of this synthetic EPO in the population, the difference between the two sex (if it is available), and the difference and the health benefits for the patients.

Response 1: Thank you for your suggestion. We added relevant information in the Discussion part (Line 615 – 621) and added three additional references 65 – 67 for supporting this information.

Round 2

Reviewer 1 Report

The revised version is suitable for publication